# Water Stress Effects on Biomass Allocation and Secondary Metabolism in CBD-Dominant *Cannabis sativa* L.

**DOI:** 10.3390/plants14081267

**Published:** 2025-04-21

**Authors:** Maddalena Cappello Fusaro, Irene Lucchetta, Stefano Bona

**Affiliations:** Department of Agronomy, Food, Natural Resources, Animals and Environment, University of Padua, 35020 Legnaro, Italy; irene.lucchetta@unipd.it

**Keywords:** *Cannabis sativa*, water stress, secondary metabolism, cannabinoids, terpenes, biomass allocation, genotype-environment interaction

## Abstract

Water availability is a key factor affecting both morphological development and secondary metabolite production in *Cannabis sativa* L. This study evaluated the effects of water stress applied during the vegetative and flowering stages on plant performance, cannabinoid concentration, and terpene composition in two Chemotype III (cannabidiol-dominant) varieties. Plants were subjected to moderate and severe water stress, and responses were assessed through biomass measurements, GC-MS analyses, and multivariate statistics. Water stress significantly influenced biomass allocation, with increased dry biomass but reduced harvest index, particularly under flowering-stage stress. Cannabidiol (CBD) content declined with increasing stress, while tetrahydrocannabinol (THC) levels increased under vegetative stress, indicating a stress-induced shift in cannabinoid biosynthesis. Cannabinol (CBN) levels also increased, suggesting enhanced THC degradation. Terpene composition was predominantly genotype-driven. PCA-MANOVA showed significant effects of variety, stress level, and their interaction, yet only minor volatiles were modulated by stress, while the most abundant terpenes remained stable across treatments, preserving the varietal aroma profile. These results underline the importance of genetic background and irrigation timing in determining cannabis yield and quality. Optimized water management is essential to ensure phytochemical consistency and sustainable production, especially in high-value medicinal and aromatic applications.

## 1. Introduction

Cannabis (*Cannabis sativa* L.), a member of the *Cannabaceae* family, is a perennial flowering herb that has long been cultivated for a wide range of applications, including food, textiles, and medicinal purposes [1].

In recent years, there has been a growing interest in cannabis, not only for its pharmacological potential but also for its economic and ecological sustainability as a crop.

The future viability of cannabis cultivation largely depends on advancements in research related to its agronomic practices. However, for decades, progress in this area was constrained by legal restrictions. With the recent easing of cannabis regulations in various parts of the world, new opportunities have emerged to explore and optimize cultivation techniques through scientific research.

Like many other crops, cannabis is susceptible to environmental stressors, with drought stress being one of the most critical challenges in the context of climate change. Drought is a significant abiotic stress that negatively impacts plant growth and agricultural productivity. It interferes with essential physiological processes such as leaf growth, enzyme activity, and photosynthetic efficiency, ultimately reducing productivity [2].

Drought stress can affect all stages of plant development, from germination to maturity. Both species-specific characteristics and environmental conditions influence the degree of impact. Plants respond to drought through complex physiological and molecular mechanisms. Central to this response is the signaling role of abscisic acid (ABA), which triggers downstream adaptations such as stomatal closure, cuticle thickening, and the synthesis of protective secondary metabolites [3,4].

During the vegetative phase, water stress significantly impacts biomass accumulation in hemp, leading to reduced plant height, leaf expansion, and stem diameter, particularly affecting fiber quality. It also alters chemical composition, decreasing cellulose content and increasing lignification. Drought can reduce hemp biomass production by up to 45%, resulting in changes to plant architecture, characterized by shorter internodes and less lateral branching [5].

The reproductive phase is highly sensitive to drought, leading to various effects such as delayed or early flowering, reduced inflorescence size and quantity, decreased pollen production in males, increased flower abortion in females, and changes in cannabinoid and terpene content [6]. Drought can affect the male-to-female plant ratio in dioecious varieties, impacting seed production. Moderate water stress during flowering may enhance bioactive compound concentrations [6]. During seed filling and maturation, water scarcity can significantly reduce seed quantity and size, alter chemical composition, and decrease seed viability and vigor. Water stress can also lower oil content in hemp seeds by 15–20% and change essential fatty acid proportions [7].

Water stress during the flowering phase of cannabis significantly alters the plant’s secondary metabolism, affecting the biosynthesis and accumulation of cannabinoids [6]. This is crucial for crops aimed at producing inflorescences rich in bioactive compounds for medical and industrial uses. Under drought conditions, hemp increases the production of protective compounds, such as volatile terpenes that lower leaf temperature [8], flavonoids with antioxidant properties, and specific secondary metabolites that enhance stress resistance [9]. These adaptations result in notable changes in the cannabinoid profile and concentration in hemp [10,11].

Moderate water stress during flowering can enhance cannabinoid concentrations in cannabis plants, particularly cannabidiol (CBD). Research has shown that maintaining substrate moisture at 60–70% of field capacity can increase CBD levels by 12–15% due to improved gene expression and a consequent greater production of trichomes [6]. However, severe water stress (below 40% field capacity) negatively affects CBD concentrations by inhibiting plant metabolism [12].

Tetrahydrocannabinol (THC) responses to water stress vary by genotype, with moderate stress potentially increasing THC levels by 8–20% and some varieties showing increases up to 25% [6]. Moderate stress also raises the THC:CBD ratio, favoring THC production [9]. In industrial hemp with low THC levels, prolonged water stress may cause THC concentrations to exceed legal limits [13].

Water stress also significantly affects the production of minor cannabinoids and terpenes in plants [14]. The terpene profile can change with an increase in volatile monoterpenes (myrcene, limonene, pinene) and enhanced sesquiterpenes (beta-caryophyllene, α-humulene) when subjected to moderate stress [15]. It must be remembered that altered relationships between terpene classes strongly affect organoleptic properties and potential pharmacological effects [16].

Understanding the impact of water availability on plant performance and phytochemical composition is crucial for improving cultivation methods, particularly in controlled-environment agriculture and medicinal cannabis production. This study aims to identify key phases where water stress most significantly affects yield and cannabinoid production, offering insights for water management to optimize water use efficiency, maintain product quality, and ensure chemical consistency.

These findings can help develop evidence-based irrigation strategies, contributing to more sustainable and resource-efficient cannabis production systems, with important implications for pharmaceutical applications and product standardization. Such responses can confer significant advantages, especially when the metabolites possess high economic value [17].

## 2. Results

### 2.1. Regression Analysis on Biomass Allocation and Cannabinoid Profiles for Stress During Vegetative Phase

Water availability, measured as the Fraction of Transpirable Soil Water (FTSW), had a notable impact on plant morphology and cannabinoid composition in *Cannabis sativa* L., with the two CBD-dominant varieties responding differently. The results are presented in Figure 1a–c.

Dry inflorescence weight did not show a significant correlation with water availability in either variety (Figure 1a), suggesting that dry yield remained relatively stable under different levels of water stress. In contrast, total dry biomass exhibited a strong positive correlation with FTSW, indicating that biomass accumulation decreased as water stress intensified in both ‘Fenomoon’ (R^2^ = 0.462, *p* < 0.001) and ‘Harlequin’ (R^2^ = 0.425, *p* < 0.001) (Figure 1b). The harvest index, representing the proportion of biomass allocated to reproductive tissues, resulted in a negative correlation with FTSW, increasing significantly with increasing water stress, particularly in ‘Fenomoon’ (R^2^ = 0.264, *p* < 0.01), suggesting that drought stress altered the plant’s biomass allocation strategy by favoring reproductive maintenance over vegetative investment (Figure 1c).

Cannabinoid concentrations also responded to water availability (Figure 2a–c). Cannabidiol (CBD) content showed a tendency to decline with increasing stress in ‘Fenomoon’ (R^2^ = 0.153, *p* < 0.05), while no significant correlation was observed in ‘Harlequin’ (Figure 2a). In contrast, both cannabinol (CBN) and tetrahydrocannabinol (THC) concentrations increased under water stress, as evidenced by their negative correlations with FTSW in both varieties. The strongest associations were found for CBN (R^2^ = 0.385 in ‘Fenomoon’, R^2^ = 0.334 in ‘Harlequin’; *p* < 0.001 for both; Figure 2b), while THC content also increased significantly under stress conditions (R^2^ = 0.202 in ‘Fenomoon’, R^2^ = 0.263 in ‘Harlequin’; Figure 2c). These patterns indicate that cannabinoid synthesis and degradation are highly sensitive to environmental conditions, which could have implications for maintaining phytochemical consistency in commercial production.

Notably, no significant Stress × Variety interactions were observed for any variable, suggesting a comparable response pattern across varieties (Table 1).

### 2.2. Regression Analysis on Biomass Allocation and Cannabinoid Profiles for Stress During Flowering Phase

When applied during the flowering phase, water stress also significantly affected morphological traits and cannabinoid profiles, although the nature and magnitude of these effects differed between the two varieties. Dry inflorescence weight decreased significantly as water stress levels increased in ‘Fenomoon’ (R^2^ = 0.553, *p* < 0.001) and to a lesser extent in ‘Harlequin’ (R^2^ = 0.188, *p* < 0.05), indicating reduced reproductive biomass accumulation under drought conditions (Figure 3a). Total dry biomass also showed a positive correlation with FTSW in both varieties (R^2^ = 0.384 in Variety A and 0.179 in Variety B, *p* < 0.001 and *p* < 0.05, respectively), confirming that vegetative growth was impaired by water deficit (Figure 3b). Harvest index showed a weaker but positive correlation with FTSW, significant in ‘Fenomoon’ (R^2^ = 0.185, *p* < 0.05) and not significant in ‘Harlequin’, suggesting a possible shift in biomass allocation under stress, though less clearly defined than in the vegetative-stage experiment (Figure 3c).

Among cannabinoids, CBD concentration showed a moderate positive correlation with FTSW in both varieties (R^2^ = 0.147 in Variety A, *p* < 0.05; R^2^ = 0.217 in Variety B, *p* < 0.01), indicating a decrease in CBD levels under increasing water stress conditions (Figure 4a). Conversely, CBN showed very weak and non-significant correlations in both varieties (Figure 4b), suggesting that water stress during flowering had limited influence on cannabinoid degradation in this stage. THC content followed a similar trend to CBD, with slightly weaker correlations (R^2^ = 0.146 in Variety A, *p* < 0.05; R^2^ = 0.127 in Variety B, *p* > 0.05), again indicating a slight reduction in THC under higher stress levels (Figure 4c).

These results confirm that morphological parameters are more responsive to water availability than cannabinoid concentrations, and that inflorescence yield is especially vulnerable to water deficits during flowering. Importantly, no significant interaction between variety and FTSW was observed, suggesting that both varieties responded in a relatively similar way to water stress at this stage (Table 2).

### 2.3. Analysis of Metabolite Profiles

To evaluate the effects of water stress and genotype on the terpene composition of *Cannabis sativa* inflorescences, a Principal Component Analysis (PCA) was first performed on the rank-transformed terpene data to reduce dimensionality. The first 27 and 24 principal components were retained for the vegetative and flowering stages, respectively, capturing at least 99% of the total variance in each case.

Multivariate analysis of variance (MANOVA) conducted on the selected principal components revealed that genotype (Variety) had a highly significant effect on overall terpene composition during both the vegetative and flowering stages (Pillai’s trace test, *p* < 0.001), confirming the strong genetic determinism of varietal aromatic profiles. The water stress level also produced a significant multivariate effect (vegetative: *p* < 0.001; flowering: *p* < 0.001), indicating that the irrigation regime modulated the terpene blend. Furthermore, a Variety × Stress level interaction was statistically significant in both stages (vegetative: *p* = 0.025; flowering: *p* = 0.003), suggesting that the response of terpene profiles to water stress is genotype-specific.

Despite these significant global effects, univariate ANOVA results suggested that the effect of water stress on the composition of volatile compounds in cannabis depends primarily on the type of compound and the phenological stage (Table 3). The major compounds, listed in bold in the following table, appear to be mainly influenced by variety, while the impact of water stress on them is minimal or even absent. This suggests that the composition of key compounds such as myrcene, limonene, α-pinene, and β-pinene is genetically determined and remains relatively stable under different environmental conditions. Among them, only caryophyllene shows a significant response to water stress, indicating that certain key molecules may still be modulated by external factors, though to a limited extent.

In contrast, the minor compounds are much more affected by water stress, in both the vegetative and reproductive phases. This suggests that stress conditions alter the plant’s secondary metabolism, either enhancing or inhibiting the synthesis of specific volatile compounds. For instance, benzaldehyde, ethylbenzene, and p-xylene exhibit notable changes under stress conditions, indicating shifts in metabolic pathways. Interestingly, for some of these compounds, the response to stress varies between the two varieties, highlighting an interaction between variety and stress level. This is particularly evident for bicyclo[2.2.1]hept-2-ene, 2,6-dimethyl-6-(4-methyl-3-pentenyl), γ-elemene, and α-caryophyllene, which show distinct patterns between the two cultivars. This indicates that the two varieties react differently to the same environmental conditions, which could have important implications for selecting varieties best suited to water-limited environments.

Differences also emerge between the vegetative and reproductive phases, with more pronounced responses to water stress observed during the reproductive stage. This is likely due to the plant’s increased metabolic investment in the synthesis of secondary metabolites at this stage, which play a crucial role in both the aroma and therapeutic properties of cannabis. Some compounds, such as borneol, copaene, and guaiol, exhibit stress-induced variations only in one of the two phases, suggesting that the regulation of volatile synthesis is closely linked to the plant’s developmental stage.

## 3. Discussion

### 3.1. Plant Growth and Biomass Allocation Under Water Stress

Water availability significantly influenced biomass dynamics, mainly when stress occurred during the flowering phase. Early drought experienced during the vegetative development stage did not notably reduce the final yield of inflorescences, indicating that plants may be able to compensate during later growth stages. This buffering effect is consistent with patterns observed in other medicinal and aromatic plants, such as basil, mint, and sage, where early stress has a lesser impact on final reproductive biomass than drought stress experienced in the later stages of growth [9,18]. When water stress occurs during the flowering stage, it directly impacts reproductive growth, resulting in a significant decrease in floral biomass, particularly in ‘Fenomoon’. This sensitivity at specific developmental stages highlights the importance of a sufficient water supply during reproduction and underscores the vulnerability of inflorescence formation to environmental challenges. These stage-specific responses are consistent with earlier studies showing that early drought primarily affects vegetative tissues due to their role in establishing photosynthetic and resource acquisition capacity [19,20].

The harvest index (HI) offers additional insight into plant allocation strategies under water stress. During vegetative-stage stress, a reduction in total biomass was observed, and upon rehydration, plants exhibited a smaller vegetative structure. This led to a proportionally higher allocation to reproductive tissues at flowering, resulting in an increased HI due to the reduction in the denominator of the HI formula (i.e., total biomass) [5,21]. In contrast, stress applied directly during flowering constrained the development of inflorescences, limiting the numerator of the index and causing HI to decline [10,11]. These findings suggest a dynamic and stage-specific shift in allocation, potentially reflecting an adaptive prioritization of reproductive effort under resource-limited conditions.

### 3.2. Cannabinoid Profiles Under Different Stress Conditions

Cannabinoid responses to water stress varied by compound and developmental stage, illustrating the complexity of the plant’s secondary metabolism. During vegetative stress, CBD content declined moderately in ‘Fenomoon’, indicating some sensitivity to early water deficits. At the same time, both THC and CBN levels increased, potentially reflecting a metabolic shift toward stress-responsive compound production. This trend is consistent with previous findings suggesting that THC accumulation may be part of the plant’s protective strategy under abiotic stress, particularly through the action of glandular trichomes that produce resinous compounds [22].

The concurrent rise in CBN under vegetative stress could indicate increased oxidative degradation of THC, as CBN is a known breakdown product. Such responses may be part of an overall stress-induced reconfiguration of the cannabinoid biosynthetic and catabolic pathways.

Under flowering-stage stress, cannabinoid modulation appeared more subdued. CBD levels continued to decline, particularly in ‘Harlequin’, while THC and CBN levels remained relatively stable. This muted response suggests that the plant may prioritize reproductive growth over secondary metabolite plasticity during this critical phase or that the stress intensity was insufficient to trigger significant changes. It may also reflect a developmental shift in metabolic sensitivity, where early-stage tissues are more biochemically responsive to water limitation.

### 3.3. Terpene Modulation Under Water Stress

The results of this study highlight the complex interplay between genotype and water availability in shaping the terpene composition of *Cannabis sativa* L. inflorescences. Multivariate analyses confirmed that genotype exerts a dominant influence on terpene profiles, reinforcing the notion that the core chemical fingerprint of each variety is genetically determined [23]. The significant effects observed for variety, water stress level, and their interaction across both the vegetative and flowering stages indicate that irrigation conditions also contribute to shaping terpene expression, although to a lesser extent than the genetic background.

Interestingly, while multivariate models detected global shifts in terpene composition, univariate analyses revealed that these differences were primarily attributable to minor compounds’ variations rather than the main aromatic constituents. This finding suggests that the biosynthesis of the most abundant terpenes, such as α-pinene, β-myrcene, and D-limonene, remains largely unaffected by water stress, maintaining a stable core aromatic profile even under varying environmental conditions. This stability is particularly relevant from an agronomic and commercial perspective, as the main compounds remain consistent in the varietal molecular profile, even when plants are subjected to moderate water deficit.

The greater plasticity of minor terpenes under stress conditions may reflect a more dynamic role for these compounds in the plant’s adaptive or defensive responses. While their concentrations are relatively low, minor volatiles often have a high olfactory impact and could contribute to subtle but perceptually relevant modifications in aroma. Such changes may also have implications for the pharmaceutical properties of cannabis, given the bioactivity attributed to several of these secondary metabolites.

Overall, the findings indicate that while water stress can modulate the fine composition of the terpene bouquet, it does not compromise the identity-defining aromatic traits of the varieties, which are robustly preserved. This suggests that carefully managed irrigation strategies can help balance water use efficiency without negatively impacting the quality attributes of the final product. From a production standpoint, understanding which components of the terpene profile are stable and which are sensitive to stress can inform cultivation practices aimed at maximizing sustainability and product uniformity.

## 4. Materials and Methods

### 4.1. Experimental Setup

The experiments were conducted in a greenhouse at the University of Padua, located in Legnaro, Italy. The trial began on 6 December 2024, using rooted cuttings with at least three true leaves from two different varieties of *Cannabis sativa* L. (Chemotype III, cannabidiol (CBD)-dominant). A total of 120 radicated cuttings, 4 weeks old, were cultivated in parallel, with plants spatially divided into two sections of the greenhouse due to logistical constraints. For this reason, the study was structured and analyzed as two independent experiments. In the first experiment, 60 cuttings were cultivated to evaluate the effects of water stress during the vegetative stage, while in the second experiment, 60 cuttings were cultivated to evaluate the effects of water stress during the flowering stage.

In each experiment, 30 cuttings of ‘Fenomoon’ (Fenocan AG, Sarnen, OW, Switzerlandand) and 30 of ‘Harlequin’ (United Cannabis Seeds, Santa Rosa, CA, USA) were transplanted into 4.5 L perforated pots containing 2 kg of a substrate composed of peat and sand (1:1 *v*/*v*). A felt pad was placed at the bottom of each pot to prevent substrate and water loss through drainage holes. The greenhouse was maintained at temperatures between 22 °C and 27 °C. Natural sunlight provided the primary source of illumination, supplemented by high-pressure sodium (HPS) lamps, which enhanced light intensity and regulated the photoperiod. Each block was equipped with two HPS lamps, ensuring an 18-h light and 6-h dark cycle, with an average supplementary PPFD of 250 µmol/m^2^/s. Plants were irrigated with water during the first eight Days After Transplant (DAT) and subsequently with the nutrient solution “CANNA Aqua Vega (solution A + B) (CANNA, Oosterhout, The Netherlands)” (EC 0.8 S/cm, pH 5.7). In both experiments, the plants were induced into flowering at 63 DAT by adjusting the photoperiod to 12 h of light and 12 h of dark and the nutrient solution was changed to a specific one for the flowering phase “CANNA Aqua Flores (solution A + B) (CANNA, Oosterhout, The Netherlands)” (EC of 1.6 S/cm and pH 5.8).

The experiment was conducted following a split-plot design with two replicates (Blocks 1 and 2). Each block was divided into two main plots, one for ‘Fenomoon’ (Variety A) and the other for ‘Harlequin’ (Variety B), with 15 plants of each variety per block. Within each main plot, the three stress levels were randomly assigned as subplots: control (C) at 90% of pot capacity, moderate stress (M) at 60% of pot capacity, and severe stress (S) at 30% of pot capacity. Each treatment included five sub-replicates (plants). Pot capacity (PC) was determined gravimetrically by saturating three representative pots with water and letting them drain overnight, resulting in a water-holding capacity of an average of 460 g per pot (In a previous experiment using the constructed substrate, the percentage of water at the permanent wilting point was determined (expressed as the percentage of water relative to the dry weight of the soil dried at 105 °C). This determination was conducted in a pot cultivation experiment with lettuce. This means that the value was not exactly the same across the three treatments, as it was not possible to analytically determine the weight at the wilting point a priori, but only at the end of the experiment). In this way, the target weight for each pot was calculated before the stress period began. All pots were irrigated to saturation and weighed. The target weight for each pot was calculated by subtracting 460 × 0.1 g for control plants, 460 × 0.4 g for plants under moderate stress, and 460 × 0.7 g for those under severe stress.

Water stress during the vegetative stage was initiated at 38 DAT. At this point, all pots were weighed at pot capacity (PC), and irrigation was withheld, allowing the pots to dry until they reached their respective target weights, corresponding to 90%, 60%, or 30% of PC. Once the target weights were reached, irrigation was adjusted to maintain the target moisture level for 10 consecutive days. Pots were weighed daily at 14:30 to monitor water loss due to evapotranspiration. After the stress period, pots were rehydrated to pot capacity and irrigated uniformly until the end of the experiment.

In the second experiment, water stress during flowering was initiated at 84 DAT. The protocol was identical to that used during the vegetative stage. Irrigation was withheld until pots reached their respective target weights for 90%, 60%, or 30% of pot capacity. Once the target weights were achieved, the target moisture levels were maintained for 10 consecutive days. After the stress period, pots were rehydrated to pot capacity and irrigated uniformly until the end of the experiment.

After a 118-day growth cycle, the inflorescences were harvested, and their fresh weight was recorded. They were then dried at 30 °C for 7 days, ensuring humidity levels were below 10%, after which their dry weight was measured. The dried inflorescences were milled. Cannabinoid and terpene contents were analyzed in the laboratory using gas chromatography-mass spectrometry (GC-MS).

At the end of the experiment, the plants without inflorescences were no longer irrigated and were left to die due to water scarcity. A plant was considered “dead” when severe wilting occurred, preventing recovery even if irrigation was resumed. The pots without aboveground biomass were also weighed to determine the actual wilting point (WP) for each pot.

Although plants were initially assigned to predefined stress levels, the statistical analysis on biomass parameters and cannabinoid content was not conducted on categorical treatments. Instead, plant responses were analyzed in relation to the actual available water for each pot, calculated as Transpirable Soil Water (TSW), following Sinclair and Ludlow (1986) [24]. TSW was determined as the difference between pot weight at pot capacity (PC), pot weight at wilting point (WP), and plant biomass. As root biomass could not be isolated from the substrate, aerial dry weight was used as a proxy. Correlation analyses were then performed to quantify the plant response to actual water availability.

In this study, we opted to use the Fraction of Transpirable Soil Water (FTSW) as our measure of soil water availability. FTSW is a normalized index of soil water content that ranges from 1 (field capacity) to 0 (wilting point) [24]. This metric was chosen for several reasons: First, FTSW provides a more plant-centric approach to water stress, as it directly relates to the amount of water available for plant transpiration [25]. Second, it allows for standardized comparisons across different soil types and plant species, as it accounts for variations in soil water holding capacity [26]. Third, FTSW has been shown to correlate well with physiological responses in plants, making it a robust indicator of plant water status [27]. Lastly, the use of FTSW enables more accurate modeling of plant responses to water stress, as it captures the non-linear nature of plant water use efficiency as soil dries [28]. These factors make FTSW a valuable tool for assessing plant-water relations in our experimental context.

### 4.2. Gas Chromatography-Mass Spectrometry Analysis

The cannabinoid and terpene contents inflorescence were analyzed in the laboratory using gas chromatography-mass spectrometry (GC-MS). To extract plant secondary metabolites (PSMs), 2 g of ground inflorescence material from each sample was placed in 20 mL of analytical-grade dichloromethane (CD_2_Cl_2_) and mixed for 30 min. The chemical composition of all samples was determined using a gas chromatography system fitted with an Agilent J&W DB-5MS column (60 m or 30 m in length, 320 μm internal diameter, and 0.50 μm film thickness). Helium served as the carrier gas at a flow rate of 1 mL/min. The GC oven temperature was initially held at 40 °C for 2 min, then increased to 160 °C at a rate of 3 °C/min, followed by a ramp of 10 °C/min to 250 °C, where it was maintained for 5 min. The separated components were analyzed by a mass spectrometer. The temperatures of the MSD transfer line, ion source, and quadrupole mass analyzer were set to 280 °C, 230 °C, and 150 °C, respectively. Ionization was performed at 70 eV, and mass detection was carried out in scan mode, covering an m/z range from 30 to 500. Data processing was completed using Mass Hunter software combined with the NIST library (Agilent Technologies, Santa Clara, CA, USA).

### 4.3. Statistical Analysis

All statistical analyses and data visualizations were conducted using R software (version 4.2.2; R Core Team, 2022) within the RStudio 4.2.2 environment. To assess the effect of water stress on plant traits and secondary metabolite production, linear regression analyses were performed using stress level (measured as a continuous variable) as the independent variable. Separate regression models were fitted for each variety to explore genotype-specific responses. The coefficient of determination (R^2^) and *p*-values were calculated for each model, and significance levels were indicated using asterisks (* *p* < 0.05; ** *p* < 0.01; *** *p* < 0.001). Additionally, an interaction term between stress and variety (Stress × Variety) was included in a separate model to test whether the response slopes significantly differed between the two varieties.

Graphical outputs were generated using ggplot2, showing regression lines, data points, R² values, and corresponding *p*-values. Summary tables of R^2^ values and statistical significance for each trait and variety are reported in the Section 2.

Regarding the terpenes analysis, due to the high dimensionality of the terpene dataset (64 compounds) relative to the sample size, a multivariate statistical approach was adopted to evaluate treatment effects. Prior to analysis, all terpene values were converted to ranks in order to reduce the influence of scale variability and non-normal distributions commonly observed in GC-MS metabolite data. This non-parametric transformation allowed for a robust assessment of relative shifts in compound abundance across treatments.

A Principal Component Analysis (PCA) was performed on the rank-transformed terpene dataset to reduce dimensionality and summarize overall variance. The PCA was conducted using the FactoMineR package in R, with standardized input data (scale.unit = TRUE) [29]. The first principal components (PCs), which collectively explained 99% of the total variance (27 for vegetative phase and 24 for flowering phase), were retained for further analysis.

Subsequently, a Multivariate Analysis of Variance (MANOVA) was applied to the selected principal components to test for the effects of Block, Variety, Stress level, and their interaction. The MANOVA was conducted using Pillai’s trace test. This approach allowed for a comprehensive evaluation of how water stress and genotype influence the overall terpene composition, while accounting for multicollinearity and dimensionality reduction.

## 5. Conclusions

This study highlights the significant impact of water stress on the morphology and phytochemical profile of *Cannabis sativa* L. Chemotype III, with responses varying according to stress intensity, growth stage, and genotype. Water limitation changed the distribution of biomass, especially during the flowering phase, reducing allocation to reproductive structures as indicated by a decline in harvest index.

Cannabinoid profiles were also affected: cannabidiol (CBD) levels declined with increasing stress. In contrast, tetrahydrocannabinol (THC) and cannabinol (CBN) levels increased under stress conditions, indicating stress-driven modulation of cannabinoid biosynthesis and degradation pathways.

Terpene composition showed more significant genetic than environmental control. Major terpenes remained largely stable under different irrigation regimes, reflecting strong varietal control [30]. Among them, only caryophyllene showed notable sensitivity to water availability. Interestingly, caryophyllene is particularly important from a therapeutic perspective due to its interaction with CBD [31]. In contrast, minor volatiles were more sensitive to water stress, particularly during the reproductive phase, and their variation was both stress- and genotype-dependent. Compounds such as benzaldehyde, ethylbenzene, and p-xylene—often associated with aromatic or phenylpropanoid pathways—may reflect stress-induced shifts in secondary metabolism [32].

These findings emphasize the need for integrated cultivation strategies that combine irrigation management with genotype selection. A better understanding of plant responses to water stress can support optimizing yield quality and chemical composition in medicinal and aromatic cannabis production.

## Figures and Tables

**Figure 1 plants-14-01267-f001:**
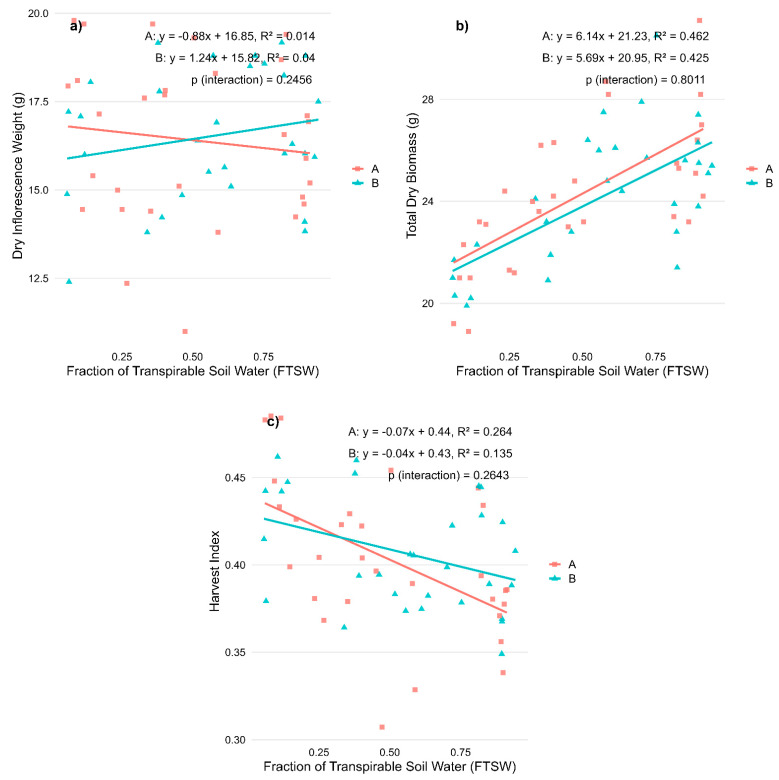
Effects of water availability (FTSW) during the vegetative stage on morphological traits in two CBD-dominant *Cannabis sativa* L. varieties. (**a**) Dry inflorescence weight; (**b**) Total dry biomass; and (**c**) Harvest index. Regression lines represent the relationship between stress levels (expressed as Fraction of Transpirable Soil Water, FTSW) and each trait for each variety (A, red squares: ‘Fenomoon’; B, blue triangles: ‘Harlequin’). *p*-values for the interaction term (Stress × Variety) are reported in each panel.

**Figure 2 plants-14-01267-f002:**
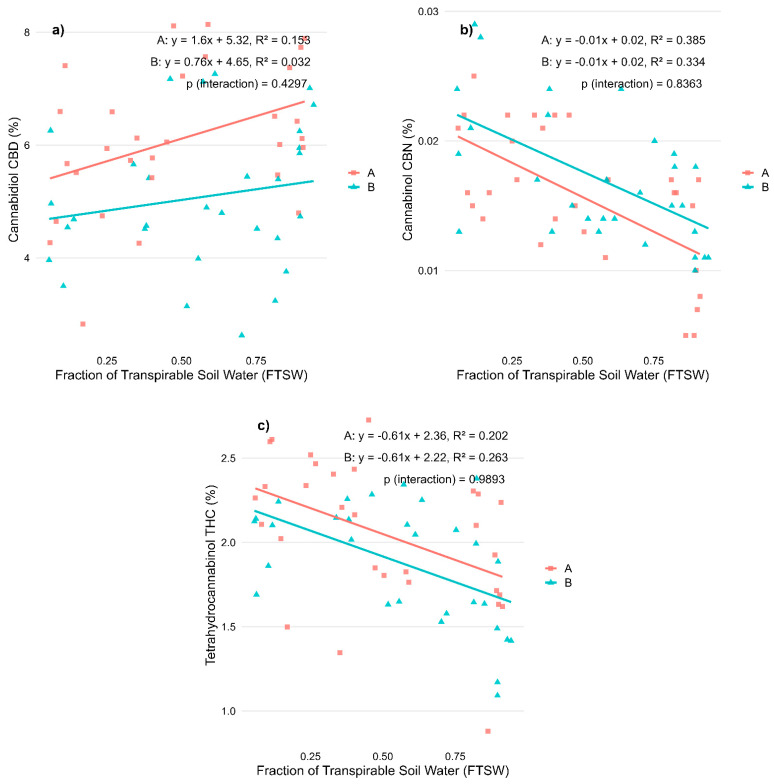
Effects of water availability (FTSW) during the vegetative stage on cannabinoid concentrations in two CBD-dominant *Cannabis sativa* L. varieties. (**a**) Cannabidiol (CBD); (**b**) Cannabinol (CBN); and (**c**) Tetrahydrocannabinol (THC) content, expressed as a percentage of inflorescence dry weight. Regression lines illustrate the relationship between stress level (FTSW) and cannabinoid concentration for each variety (A, red squares: ‘Fenomoon’; B, blue triangles: ‘Harlequin’). *p*-values for the interaction term (Stress × Variety) are indicated in each plot.

**Figure 3 plants-14-01267-f003:**
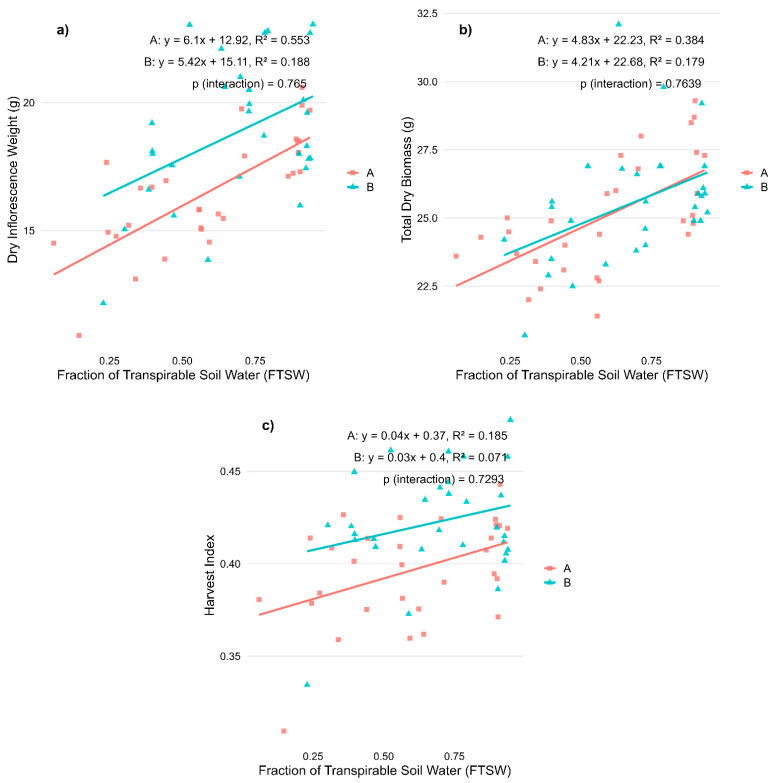
Effects of water availability (FTSW) during the flowering stage on morphological traits in two CBD-dominant *Cannabis sativa* L. varieties. (**a**) Dry inflorescence weight; (**b**) Total dry biomass; and (**c**) Harvest index. Regression lines represent the relationship between stress levels (expressed as Fraction of Transpirable Soil Water, FTSW) and each trait for each variety (A, red squares: ‘Fenomoon’; B, blue triangles: ‘Harlequin’). *p*-values for the interaction term (Stress × Variety) are reported in each panel.

**Figure 4 plants-14-01267-f004:**
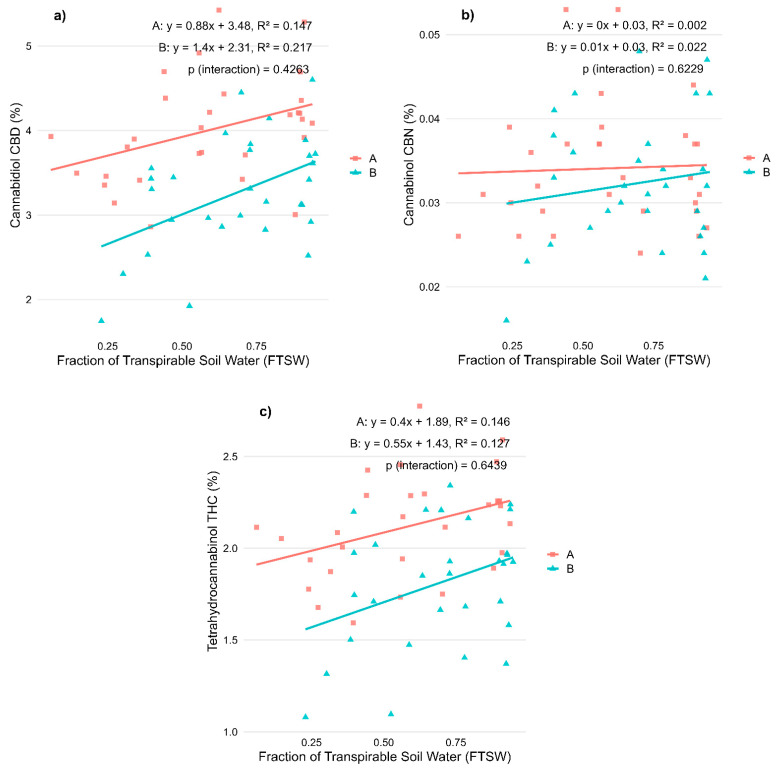
Effects of water availability (FTSW) during the flowering stage on cannabinoid concentrations in two CBD-dominant *Cannabis sativa* L. varieties. (**a**) Cannabidiol (CBD); (**b**) Cannabinol (CBN); and (**c**) Tetrahydrocannabinol (THC) content, expressed as a percentage of inflorescence dry weight. Regression lines illustrate the relationship between stress level (FTSW) and cannabinoid concentration for each variety (A, red squares: ‘Fenomoon’; B, blue triangles: ‘Harlequin’). *p*-values for the interaction term (Stress × Variety) are indicated in each plot.

**Table 1 plants-14-01267-t001:** Summary of linear regression analyses between stress levels (FTSW) and plant morphological and chemical parameters in two CBD-dominant *Cannabis sativa* L. varieties during the vegetative stage. The table reports the coefficient of determination (R²) and corresponding significance levels for each variety (A: ‘Fenomoon’; B: ‘Harlequin’), as well as the *p*-value for the interaction between Stress and Variety (Stress × Variety). Asterisks indicate significance levels of the regression models: * *p* < 0.05, ** *p* < 0.01, *** *p* < 0.001.

Variable	R^2^—A	*p*-Value—A	R^2^—B	*p*-Value—B	*p*-Value—Interaction	R^2^—Total	*p*-Value—Total
Dry weight inflorescences (g)	0.014		0.040			0.001	
Dry weight biomass (g)	0.462	***	0.425	***		0.435	***
Harvest index	0.264	**	0.135	*		0.194	***
CBD (%)	0.153	*	0.032			0.047	
CBN (%)	0.385	***	0.334	***		0.332	***
THC (%)	0.202	*	0.263	**		0.239	***

**Table 2 plants-14-01267-t002:** Summary of regression analysis between water availability and agronomic and phytochemical parameters under flowering-stage water stress. R^2^ values and corresponding significance levels (* *p* < 0.05; ** *p* < 0.01; *** *p* < 0.001) are reported for each variable and variety (A and B), along with the *p*-value for the interaction term (Stress × Variety). The table also includes overall R^2^ and *p*-values calculated without distinction between varieties. The variables include yield components and cannabinoid percentages measured at harvest.

Variable	R^2^—A	*p*-Value—A	R^2^—B	*p*-Value—B	*p*-Value—Interaction	R^2^—Total	*p*-Value—Total
Dry weight inflorescences (g)	0.553	***	0.188	*		0.348	***
Dry weight biomass (g)	0.384	***	0.179	*		0.283	***
Harvest index	0.185	*	0.071			0.156	**
CBD (%)	0.147	*	0.217	**		0.067	*
CBN (%)	0.002		0.022			0.004	
THC (%)	0.146	*	0.127			0.052	

**Table 3 plants-14-01267-t003:** Statistical significance of the effects of variety (V), water stress (S), and their interaction (V × S) on volatile compounds identified in cannabis inflorescences by GC-MS analysis. The *p*-values are reported for each compound. The main compounds are listed in bold. Significance levels are indicated as follows: *** *p* < 0.001, ** *p* < 0.01, * *p* < 0.05, and “.” for *p* < 0.1.

	Compound	Vegetative			Flowering		
		Variety	Stress	V × S	Variety	Stress	V × S
1.	Dimethyl sulfide	**	.				
2.	Propanal, 2-methyl-	***			**		
3.	2,3-Butanedione	**	*				
4.	Acetic acid, methyl ester	***					
5.	Acetic acid	***			**		
6.	Furan, 3-methyl-	***			***		
7.	Butanal, 2-methyl-	**			*	.	
8.	Propanoic acid	*					
9.	Furan, 2-ethyl-	***			**	*	
10.	2-Butanone, 3-hydroxy-	***			**		
11.	1-Butanol, 2-methyl-		*				
12.	Butanoic acid	**			**	.	
13.	2,3-Butanediol	***			***	.	
14.	Hexanal	*					
15.	3-Hexen-1-ol, (E)-				**		
16.	2-Hexenal	**					
17.	Ethylbenzene	***			**	**	
18.	1-Hexanol		.		*		
19.	p-Xylene	***			**	*	
20.	Heptanal	***			***		
21.	Bicyclo[2.2.1]hept-2-ene, 2,7,7-trimethyl-	***	**		***		
**22.**	**.alpha.-Pinene**	*******			*******		
**23.**	**Camphene**	*******			******		
24.	Benzaldehyde	***	.	*	**		
25.	.beta.-Pinene Phellandreene	***			***		
**26.**	**Myrcene**	*******			*******		
**27.**	**Acetic acid, hexyl ester**	*******			*******		
28.	Benzene, 1-methyl-3-(1-methylethyl)-		***				
**29.**	**D-Limonene**	*******	**.**		*******	**.**	*****
30.	1,3,6-Octatriene, 3,7-dimethyl-, (E)-	***	.		***		***
31.	1,3,6-Octatriene, 3,7-dimethyl-, (Z)-	***		*	***	*	
32.	1,4-Cyclohexadiene, 1-methyl-4-(1-methylethyl)-	***			***		
33.	1,6-Octadien-3-ol, 3,7-dimethyl-	**		**	***		
34.	Phenylethyl Alcohol	***			***	**	
35.	trans-2-Pinanol	***			***		
36.	2,4,6-Octatriene, 2,6-dimethyl-	***	.		***		
37.	Bicyclo[3.1.1]heptan-3-ol, 6,6-dimethyl-2-methylene-	***		.		*	*
38.	2,7-Octadien-4-ol, 2-methyl-6-methylene-, (S)-	***			***	*	
**39.**	**Borneol**	*******			*******	*****	
40.	3-Cyclohexen-1-ol, 4-methyl-1-(1-methylethyl)-, (R)-				*		
41.	3-Cyclohexene-1-methanol, .alpha., .alpha.4-trimethyl-			*	***	***	*
42.	6-Octen-1-ol, 3,7-dimethyl-, (R)-	***		*	***		
43.	Benzeneacetic acid, ethyl ester	*			***	.	
44.	Bicyclo[2.2.1]heptan-2-ol, 1,7,7-trimethyl-, acetate, (1S-endo)-	***			***		
45.	3a,7-Methano-3aH-cyclopentacyclooctene, 1,4,5,6,7,8,9,9a-octahydro-1,1,7-trimethyl-, [3aR-(3a.alpha.,7.alpha.,9a.beta.)]-	***	**		***		
46.	Ylangene	***			***		
47.	Copaene	***	*		***		
48.	Hexanoic acid, hexyl ester	***	*		***		
**49.**	**Caryophyllene**	*******	*****	**.**	*******		
50.	Bicyclo[3.1.1]hept-2-ene, 2,6-dimethyl-6-(4-methyl-3-pentenyl)-	***	.		***		
51.	.alpha.-Caryophyllene	***	.	*	***		
52.	1,6,10-Dodecatriene, 7,11-dimethyl-3-methylene-, (E)-	***			***		
53.	Naphthalene, 1,2,3,4,4a,5,6,8a-octahydro-7-methyl-4-methylene-1-(1-methylethyl)-, (1.alpha.,4a.alpha.,8a.alpha.)-	***	*				
54.	.gamma.-Elemene	***	.	*			
55.	Naphthalene, 1,2,4a,5,6,8a-hexahydro-4,7-dimethyl-1-(1-methylethyl)-, (1.alpha.,4a.alpha.,8a.alpha.)-						
56.	.alpha.-Calacorene	***	*				
57.	1H-Cycloprop[e]azulene, decahydro-1,1,7-trimethyl-4-methylene-, [1aR-(1a.alpha.,4a.alpha.,7.alpha.,7a.beta.,7b.alpha.)]-	*					
58.	Caryophyllene oxide		*				
59.	Guaiol	***					
60.	2-Naphthalenemethanol, 1,2,3,4,4a,5,6,7-octahydro-.alpha.,.alpha.,4a,8-tetramethyl-, (2R-cis)-	***					
61.	2-Naphthalenemethanol, decahydro-.alpha.,.alpha.,4a-trimethyl-8-methylene-, [2R-(2.alpha.,4a.alpha.,8a.beta.)]-	***					
62.	2-Naphthalenemethanol, 1,2,3,4,4a,5,6,8a-octahydro-.alpha.,.alpha.,4a,8-tetramethyl-, [2R-(2.alpha.,4a.alpha.,8a.beta.)]-	***	*				
63.	5-Azulenemethanol, 1,2,3,3a,4,5,6,7-octahydro-.alpha.,.alpha.,3,8-tetramethyl-, [3S-(3.alpha.,3a.beta.,5.alpha.)]-	***					
**64.**	**.alpha.-Bisabolol**	*******					

## Data Availability

The original data used for the analysis are present in the Appendix A. Further inquiries can be directed to the corresponding authors.

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
