# Peer review of "Water Stress Effects on Biomass Allocation and Secondary Metabolism in CBD-Dominant Cannabis sativa L."

_plants, 2025, doi:10.3390/plants14081267_

Round 1
Reviewer 1 Report
Comments and Suggestions for Authors
The manuscript explored the effects water stress on Cannabis sativa L. from biomass allocation and secondary metabolism, analyzed the changes of these indexes at the vegetative and flowering stages. Biomass allocation was mainly affected by water stress, while secondary metabolism, particularly terpenes, was predominantly genotype-driven. The results in this manuscript present some important clues for irrigation timing for the cannabis yield and quality and optimized water management. However, there were some problems within it.
- In the Introduction, the logical flowshowed relative weak, such as lines 31-41, 44-49.
- In the Discussionsection, there are many sentences were repeated with the relative results, for example lines 273-277.
- In the Results, the description of some sentences seemed single and repetitive, such as lines 104-107 with lines 160-162.
- The conclusion, it showed longer and repetitive with the Results.
- The title of 2.2 is same with 2.1of Results? Should be flowering phase?
- The Table 3, the yellow parts should be Black body?
- The language presented some discomfort to readers.
Author Response
Dear Reviewer,
We sincerely thank you for your thoughtful and constructive feedback on our manuscript entitled “Water stress effects on biomass allocation and secondary metabolism in CBD-Dominant Cannabis sativa L.” (Manuscript ID: plants-3568913), submitted to Plants for the Special Issue “Cannabis sativa: Advances in Biology and Cultivation—2nd Edition.”
Your insights have helped us significantly improve our manuscript's clarity, structure, and scientific depth. Below, we provide a detailed point-by-point response to each of your suggestions. In the revised manuscript:
- A thorough reworking of the Discussion and Conclusion section to reduce redundancy and deepen interpretation. Repetitive statistical statements were removed, and the text now focuses on contextualising findings concerning existing literature.
- Stylistic edits to the Results section, particularly in sections 2.1 and 2.2, where introductory sentences were rephrased to avoid repetitive phrasing between the vegetative and flowering stages.
- Following a helpful comment from one of the reviewers regarding the naming of a variable in one of the figures, we decided to revise the axis labels across all graphs to improve clarity and visual consistency. Additionally, we removed the variable names in brackets from the main text, aiming to enhance the overall readability and flow. We hope these adjustments contribute to a clearer and more professional presentation of the data.
Changes in the main manuscript were made using the “track changes” features in Microsoft Word.
Please find below a detailed point-by-point response to your comments. We hope the improvements meet your expectations and that the manuscript is now suitable for publication in Plants.
We remain at your disposal for any further clarification.
Comment 1: In the Introduction, the logical flowshowed relative weak, such as lines 31-41, 44-49.
Response 1: Thank you for your comment regarding the logical flow of the Introduction. We have revised this section to improve coherence and readability, while preserving the key content and context. We hope the updated version provides a clearer and more structured narrative. Please let us know if further clarification is needed.
Comment 2: In the Discussionsection, there are many sentences were repeated with the relative results, for example lines 273-277.
Response 2: Thank you for pointing this out. The Discussion section has been thoroughly revised to minimise repetition of the Results section. Instead of repeating statistical details, the new version focuses on interpretation, contextualisation with relevant literature, and highlighting the broader implications of the findings. We hope this improves both the clarity and depth of the discussion.
Comment 3: In the Results, the description of some sentences seemed single and repetitive, such as lines 104-107 with lines 160-162.
Response 3: Thank you for highlighting this redundancy. The introductory sentences at the beginning of each Results subsection have been revised to reduce repetition and improve stylistic variety, while still clearly introducing the specific context (vegetative vs. flowering phase). The updated phrasing maintains clarity but avoids using nearly identical wording across sections.
Comment 4: The conclusion, it showed longer and repetitive with the Results.
Response 4: Thank you for your comment. We agree and have revised the Conclusions section accordingly, condensing it into a brief, more concise paragraph.
Comment 5: The title of 2.2 is same with 2.1of Results? Should be flowering phase?
Response 5: Thank you for pointing this out. You are correct; the title should refer to the flowering phase. We have corrected the section title accordingly.
Comment 6: The Table 3, the yellow parts should be Black body?
Response 6: Thank you for your observation. We initially highlighted specific components in yellow to emphasise the most relevant ones. However, we understand your concern and agree that a more standard formatting approach is appropriate. Therefore, we have replaced the yellow highlighting with bold font to improve clarity and consistency.
Additionally, we have added a column to Table 3 with numbers for each row, which we hope will make the table more readable and easier to follow. If you think it's useful, we could also reference these major compounds in the text (line 231) by stating, “The major compounds, bolded as numbers 22, 23, 26, 27, 29, 39, 49, 51, and 64 in the table below, [...]”.
Comment 7: The language presented some discomfort to readers.
Response 7: Thank you for your feedback regarding the language. We have thoroughly revised the manuscript for clarity, style, and fluency. In addition to addressing structural and content-related comments, we carefully reworded several sections to improve readability and overall coherence. We hope that the modifications made have also contributed to a higher standard of English throughout the manuscript.
Reviewer 2 Report
Comments and Suggestions for Authors
This manuscript evaluated the effects of water stress applied during the vegetative and flowering stages on plant performance, cannabinoid concentration, and terpene composition in two Chemotype III varieties. The results underlined the importance of genetic background and irrigation timing in determining cannabis yield and quality. Authors concluded that optimized water management is essential to ensure phytochemical consistency and sustainable production, especially in high-value medicinal and aromatic applications. In general, this MS provided sufficient background, designed appropriately, adequately described the methods, presented the results clearly, as well as the results supported the conclusions basically. Some details should be paid attention as follows:
- in abstract: CBD, CBN, THC should be presented by full name.
- line 108: Dry inflorescence weight (DW_Inflorescences_g) should be Dry inflorescence weight (DW/g) or (DW·g-1)? The same as other places and figs.
- line 111: Fraction of Transpirable Soil Water (FTSW) should present only FTSW because in line 105 it already occurred.
- line 355: Cannabis sativa should be italic.
- in my opinion, Conclusions is tedious and needs to be condensed to one paragraph.
- pay attention the references’ format especially ref. 10.
Author Response
Dear Reviewer,
We sincerely thank you for your thoughtful and constructive feedback on our manuscript entitled “Water stress effects on biomass allocation and secondary metabolism in CBD-Dominant Cannabis sativa L.” (Manuscript ID: plants-3568913), submitted to Plants for the Special Issue “Cannabis sativa: Advances in Biology and Cultivation—2nd Edition.”
Your insights have helped us significantly improve our manuscript's clarity, structure, and scientific depth. Below, we provide a detailed point-by-point response to each of your suggestions. In the revised manuscript:
- A thorough reworking of the Discussion and Conclusion section to reduce redundancy and deepen interpretation. Repetitive statistical statements were removed, and the text now focuses on contextualising findings concerning existing literature.
- Stylistic edits to the Results section, particularly in sections 2.1 and 2.2, where introductory sentences were rephrased to avoid repetitive phrasing between the vegetative and flowering stages.
- Following a helpful comment from one of the reviewers regarding the naming of a variable in one of the figures, we decided to revise the axis labels across all graphs to improve clarity and visual consistency. Additionally, we removed the variable names in brackets from the main text, aiming to enhance the overall readability and flow. We hope these adjustments contribute to a clearer and more professional presentation of the data.
Changes in the main manuscript were made using the “track changes” features in Microsoft Word.
Please find below a detailed point-by-point response to your comments. We hope the improvements meet your expectations and that the manuscript is now suitable for publication in Plants.
We remain at your disposal for any further clarification.
Comment 1: in abstract: CBD, CBN, THC should be presented by full name.
Response 1: Thank you for pointing this out. We agree with this comment. Therefore, we have added your suggestion.
Comment 2: line 108: Dry inflorescence weight (DW_Inflorescences_g) should be Dry inflorescence weight (DW/g) or (DW·g-1)? The same as other places and figs.
Response 2: Thank you for your comment. We understand that the expression “(DW_Inflorescences_g)” may have caused confusion. This was originally included to reflect the exact variable names used in our dataset and not a unit of measurement. It was intended to ensure consistency across the text and figures. However, we now recognize that this formatting may appear unclear or overly technical.
To address this, we have revised the axis labels across all figures for improved clarity and visual coherence. We have also removed the dataset-style variable names from the main text, with the aim of enhancing readability and creating a more fluid and polished manuscript. We hope these changes meet your expectations and improve the overall presentation of the work.
Comment 3: line 111: Fraction of Transpirable Soil Water (FTSW) should present only FTSW because in line 105 it already occurred.
Response 3: Thank you for pointing this out. We agree with the comment and have removed the repeated definition of the acronym.
Comment 4: line 355: Cannabis sativa should be italic.
Response 4: Thank you for your comment. Cannabis sativa has been italicised as suggested.
Comment 5: in my opinion, Conclusions is tedious and needs to be condensed to one paragraph.
Response 5: Thank you for your suggestion. We agree with your comment and have revised the Conclusions section accordingly, condensing it into a brief, more concise paragraph.
Comment 6: pay attention the references’ format especially ref. 10.
Response 6: Thank you for your observation. We have carefully reviewed and corrected the formatting of all references, including Reference 10, to ensure consistency with the journal's style guidelines.
Reviewer 3 Report
Comments and Suggestions for Authors
The study reported different effects of water stress on Cannabis sativa plant development, flowering, and contents of major secondary metabolites, and terpenes that influence medicinal quality of the plants. Experiments and data are reasonable and results are quite useful and somewhat significant to Cannabis cultivation. The data interpretation and conclusions are scientifically meaningful.
Author Response
Dear Reviewer,
We thank you for the positive evaluation and for taking the time to read our manuscript entitled “Water stress effects on biomass allocation and secondary metabolism in CBD-Dominant Cannabis sativa L.” (Manuscript ID: plants-3568913).
We would like to inform you that the revised version of the manuscript includes substantial improvements based on the feedback received from the other reviewers. These modifications involve:
- A restructured and refined Discussion section, with reduced repetition and greater focus on interpretation.
- Stylistic and phrasing adjustments in the Results section to improve flow and avoid redundancy.
- A comprehensive language revision to enhance clarity and readability throughout the manuscript.
Changes in the main manuscript were made using the “track changes” features in Microsoft Word.
We hope the revised version meets your expectations and sincerely thank you again for contributing to the review process.